# Neural Latents Benchmark '21: Evaluating latent variable models of neural population activity

**Felix Pei**[1*], **Joel Ye**[1,2*], **David Zoltowski**[4], **Anqi Wu**[1,5],
**Raeed H. Chowdhury**[6], **Hansem Sohn**[7], **Joseph E. O'Doherty**[8], **Krishna V. Shenoy**[9],
**Matthew T. Kaufman**[10], **Mark Churchland**[5], **Mehrdad Jazayeri**[7], **Lee E. Miller**[11],
**Jonathan Pillow**[4], **Il Memming Park**[12], **Eva L. Dyer**[1,3], **Chethan Pandarinath**[1,3†]

[1]Georgia Institute of Technology,[2]Carnegie Mellon University, [3]Emory University,
[4]Princeton University,[5]Columbia University,[6]University of Pittsburgh,
[7]Massachusetts Institute of Technology,[8]Neuralink Corp.,[9]Stanford University,
[10]University of Chicago,[11]Northwestern University, [12]Stony Brook University

## Abstract

Advances in neural recording present increasing opportunities to study neural activity in unprecedented detail. Latent variable models (LVMs) are promising tools for analyzing this rich activity across diverse neural systems and behaviors, as LVMs do not depend on known relationships between the activity and external experimental variables. However, progress with LVMs for neuronal population activity is currently impeded by a lack of standardization, resulting in methods being developed and compared in an ad hoc manner. To coordinate these modeling efforts, we introduce a benchmark suite for latent variable modeling of neural population activity. We curate four datasets of neural spiking activity from cognitive, sensory, and motor areas to promote models that apply to the wide variety of activity seen across these areas. We identify unsupervised evaluation as a common framework for evaluating models across datasets, and apply several baselines that demonstrate benchmark diversity. We release this benchmark through EvalAI. http://neurallatents.github.io

## 1   Introduction

A central pursuit of neuroscience is to understand how the rich sensory, motor, and cognitive functions of the brain arise from the collective activity of populations of neurons. To this end, we are witnessing a sea change in systems neuroscience, as a decade of rapid progress in neural interfacing technologies has begun to enable access to the simultaneous activity of vast neuronal populations [1]. As a result, neuroscientists increasingly capture high-dimensional and dynamic portraits of activity from a variety of brain areas and during diverse behaviors. The resulting datasets may stymie traditional analytical approaches that were designed around recordings from one or a handful of neurons at a time.

In response to the increased data complexity, computational neuroscientists are producing powerful methods for uncovering and interpreting structure from neural population activity. An emerging and particularly promising set of approaches – termed *latent variable models* (LVMs) – characterizes patterns of covariation across a neuronal population to reveal its internal state [2]. LVMs have proven useful for summarizing and visualizing population activity, relating activity to behavior, and interrogating the dynamic mechanisms that mediate population-level computations [3–7].

---

*Equal contribution.
†Correspondence to chethan@gatech.edu

35th Conference on Neural Information Processing Systems (NeurIPS 2021) Track on Datasets and Benchmarks.

A key opportunity to advance the application of LVMs to neural data is to capitalize on the dramatic advances in machine learning over the past decade. Yet there is a high barrier to entry for ML experts to make an impact, likely stemming from the lack of standardized datasets and metrics for evaluating and comparing LVMs. To address this gap, we introduce the **Neural Latents Benchmark** (NLB), a series of benchmark suites that will enable standardized quantitative evaluation of LVMs on neural data. These suites will provide curated datasets, standardized APIs, and example codepacks.

Here we present our first suite, **NLB '21**, which aims to broaden the potential applicability of LVM approaches by benchmarking unsupervised modeling on datasets from a variety of brain regions, behaviors, and dataset sizes. While LVMs are often developed and evaluated using data from a single brain region and behavior, activity from different regions or behaviors may have markedly different structure and thus present different modeling challenges [8]. NLB '21 provides curated neurophysiological datasets from monkeys that span motor, sensory, and cognitive brain regions, with behaviors that vary from pre-planned, stereotyped movements to those in which sensory input must be dynamically integrated and acted upon. Further, because neuroscientific datasets can vary substantially in size and different LVMs may be more or less data-efficient, we introduce data scaling benchmarks that evaluate LVM performance on datasets of different sizes. To achieve general evaluation of LVMs regardless of brain region, behavior, or dataset size, we adopt a standardized unsupervised evaluation metric known as *co-smoothing* [9], which evaluates models based on their ability to predict held-out neuronal activity. We also provide secondary evaluation metrics whose applicability and utility vary across datasets. To ensure accessibility and standardization, we provide datasets in the Neurodata Without Borders format [10], host the benchmarks and leaderboards on the EvalAI platform [11], and offer a code package that demonstrates data preprocessing and submission. To ensure consistent evaluation and to mitigate issues with overfitting or hyperparameter hacking, model predictions are evaluated against private data that are unavailable to developers.

This manuscript first motivates the broad use of LVMs for interpreting neuroscientific data. We then detail benchmark datasets and evaluation metrics, and provide examples of their application with a variety of current LVM approaches. We anticipate that this benchmark suite will provide valuable points of comparison for LVM developers and users, enabling the community to identify and build upon promising approaches.

## 1.1 Scientific Motivation and Evaluation Philosophy

LVMs are a powerful approach for characterizing the internal state of biological neural networks based on partial observations of the neuronal population. In applications to neuronal population activity, LVMs are generative models that describe observed activity as a combination of latent variables, which are often fewer than the number of observed neurons and typically exhibit an orderly progression in time. LVM approaches to neural data are grounded in empirical findings that neurons in large neuronal populations do not act independently, but rather exhibit coordinated fluctuations [12–14]. We point the reader to a recent review [2] for discussion on how LVMs can be used for neuroscientific insights. In NLB '21, we focus on unsupervised LVMs that are not directly conditioned on measured external variables. Without a strict dependence on external variables, such models have broad applicability, including settings where we cannot observe or even know the relevant external variables that affect a neural population's response.

To understand the utility of LVMs in probing neural circuits, a helpful analogy is the task of reverse engineering an artificial network that takes in a set of inputs, performs some computation, and produces a set of outputs. Between input and output, the artificial network has a set of intermediate representations, and studying these representations provides insight into the computational strategy employed by the network. Likewise, neural population recordings are observations of intermediate representations of a biological neural network that processes information and coordinates behavior.

However, the task of understanding these representations may be considerably more challenging for neural population recordings than for artificial networks, as we typically have recordings from a limited number of neurons in one or a few brain regions, we may not know anatomical connectivity, and there may be many steps of processing between externally-measurable inputs and outputs and the brain area(s) under study. Further, the observed neuronal responses are highly variable and seemingly noisy [3], making the task of understanding neural population activity inherently statistical and well-suited for latent variable approaches.

There are many potential ways to model neural population activity with latent variables, and different assumptions can lead to varied model structures that are all seemingly "correct." Given this ill-posed nature of latent variable modeling, we sought a primary evaluation approach that was largely agnostic to the form and structure of the LVM being evaluated. Indeed, the co-smoothing evaluation (defined in Section 3.2) is an unsupervised approach that only assesses the ability of LVMs to describe the observed neuronal activity itself. This allows for flexibility in modeling assumptions, while avoiding the intricate complexity of comparing vastly varying structures of LVMs.

## 1.2 Related Work

**Evaluation strategies for LVMs applied to neural data** Many new LVMs have been developed to meet the need to analyze large scale neural population recordings. For context, we document appearances of such neural data LVMs in ML venues in supplementary Table 3. These LVMs are evaluated on a variety of datasets which collectively span many areas of the brain. Other animal models (rats [15], mice [16]), and non-electrophysiological recording modalities (calcium imaging [16], fMRI [17]) are used as well. The diversity of previously used datasets is impressive, but the lack of standardized datasets has made comparison across models difficult. Moreover, even when two LVMs use the same dataset, they will often report on different metrics, or different variations of the same metric, as evidenced in Table 3. Non-standardized evaluation has made it extremely difficult to track the state of the field.

**Existing benchmarks in computational neuroscience**. The computational neuroscience community has recently produced several benchmarks around the interaction of machine learning methods and neural data. Some focus on the challenge of extracting spiking activity (action potentials, or correlates) from raw neurophysiological data, for example, the spikefinder challenge [18] for inferring spiking activity from two-photon calcium imaging data, and SpikeForest [19] for extracting spikes from electrophysiological recordings. Other benchmarks evaluate single neuron modeling to predict spike times [20] and decoding externally-measurable variables from neural population activity in motor and somatosensory cortices and hippocampus [21]. Separately, Brain-Score evaluates the alignment of deep neural networks trained to perform behavioral tasks and the brain areas associated with those tasks (particularly the ventral visual stream) [22]. Distinct from all of these, NLB '21 quantifies how well LVMs can describe neuronal population activity in an unsupervised manner.

**Evaluating generative model latents and outputs**. A candidate approach for comparing LVMs is to evaluate the quality of the model's intermediate representations. In ML, such intermediate latent representations are typically assessed via supervised evaluation, *e.g.* by measuring their disentanglement relative to data categories [23] or transfer performance in a variety of downstream tasks [24]. As described earlier, however, supervised evaluation may be limiting in some neuroscience applications, where external variables may be unreliable or incomplete descriptions of neuronal activity (particularly in higher order brain areas, such as those underlying cognition). Therefore, achieving performance metrics that generalize across brain regions and task conditions requires unsupervised evaluation.

A common approach to unsupervised evaluation of latent representations is to assess the quality of a model's output, via one-to-one comparisons with reference data. This can be achieved via prediction of held-out portions of the data, as in MLM (masked language modeling) [25] or image inpainting [26]. When used as a learning objective, these metrics can risk "shortcut" solutions based on shallow data correlations [26]. On the other hand, MLM approaches have been demonstrably effective at inducing high-level, semantic representations [25], justifying their use as a first proxy objective in absence of satisfactory supervised metrics.

## 2 NLB '21: Datasets

To facilitate the development of LVMs for broad application, our datasets span brain regions that underlie motor, sensory, and cognitive functions, and span a variety of behaviors. As described below, each dataset presents a unique set of challenges to uncovering neural latents. We note that while all datasets have been used in previous studies (primarily for neuroscientific purposes), there have not been any previous benchmarks for neural data LVM evaluation, either using these data, or any other that we know of.

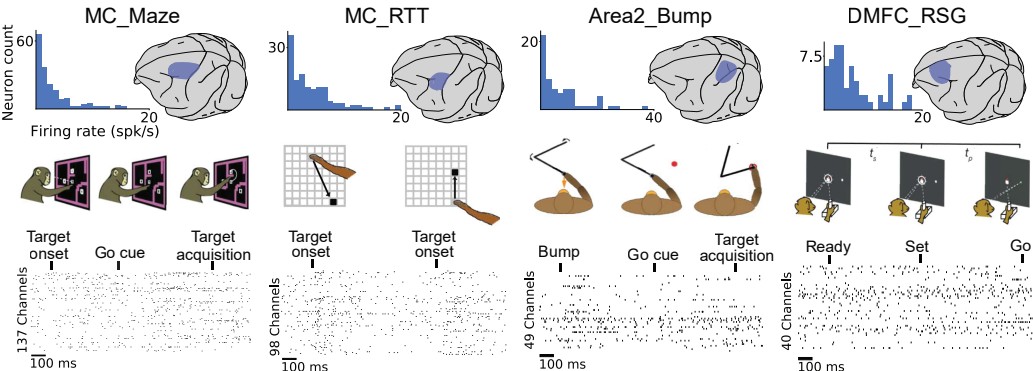

Figure 1: **NLB '21 datasets span four diverse brain area/task combinations**. For each behavioral task (center row), the top left panel presents the distribution of firing rates of the neurons in the dataset, while the top right highlights the recorded brain area. Lower panels present sample spike rasters, aligned to task events. **Tasks:** Motor cortical (MC) datasets include a center-out instructed delay reaching task with stereotyped conditions (Maze) and a continuous series of reaches in a random target task (RTT). Data from somatosensory cortex (Area 2) include externally-perturbed movements and volitional, goal-directed movements. Data from dorso-medial frontal cortex (DMFC) are during the Ready-Set-Go (RSG) cognitive time interval reproduction task.

All datasets contain electrophysiological measurements recorded using intracortical microelectrode arrays. Preliminary signal processing was applied to the raw voltage recordings to extract spiking activity. While this process of spike sorting is imperfect and a subject of active study, we view it as a distinct problem from the process of extracting latent structure in the data, and thus provide datasets in spike-sorted form. Detailed descriptions of each dataset can be found in the Appendix. All datasets are available through DANDI (Distributed Archives for Neurophysiology Data Integration) in the NWB (Neurodata without Borders) standard [27]. Links can be found at https://neurallatents.github.io/. Note that our benchmark uses select recording sessions from larger datasets with potentially many other sessions. Thus, related data to those we present here may have been separately uploaded for public use (e.g., on DANDI), but those releases exclude the specific recording sessions used for NLB'21.

**MC_Maze.** The Maze datasets consist of recordings from primary motor and dorsal premotor cortices while a monkey made reaches with an instructed delay to visually presented targets while avoiding the boundaries of a virtual maze [28]. The monkey made reaches in 108 behavioral task configurations, where each task configuration used a different combination of target position, numbers of virtual barriers, and barrier positions. These different configurations elicited a wide variety of straight and curved reach trajectories, and each configuration was attempted by the monkey many times in random order, resulting in thousands of trials recorded across a given experimental session.

The Maze datasets are exceptional in their combination of behavioral richness (number of task configurations), stereotyped behavior across repeated trials (tens of repeats for each task configuration), and high total trial counts (thousands) – these attributes support averaging neuronal activity across repeated trials as a simple, first-pass de-noising strategy [3], while retaining enough diversity in task conditions to allow rich investigation into the structure of the population activity [29]. Additionally, the instructed delay paradigm allows movement preparation before presenting a go cue, which enables a clean separation of the neural processes related to preparation and execution [28, 30]. Due to the instructed delay paradigm and lack of unpredictable task events, population activity during the execution phase is largely predictable based on the state of the neural population at preparation, creating a unique case where activity can be well-modeled as an autonomous dynamical system [30–32, 4]. With their unique properties, the Maze datasets have been extensively used in neuroscientific studies - in particular, they have been critical for revealing a plethora of insights into the structure of neural population activity during movement preparation and execution [28, 31, 33–40]. They have also been used for validating a few LVMs individually [41, 32, 42, 8].

MC_Maze consists of one full session with 2869 total trials and 182 neurons, with simultaneously monitored hand kinematics. We expect this dataset to serve as a basic yet versatile baseline for LVM development (akin to a "neuroscience MNIST"). Additionally, to support a data-scaling benchmark that characterizes data efficiency of LVMs, we provide three scaled datasets, each from separate

experimental sessions, containing 500, 250, and 100 training trials and 100 test trials each. We refer to the scaled versions of this dataset as `MC_Maze-L`, `MC_Maze-M`, `MC_Maze-S`, respectively. Each scaled dataset contains only 27 conditions and neuron counts ranging from 142 to 162. A number of other Maze datasets are available on DANDI [43].

**MC_RTT.** Though the Maze datasets contain rich behavior, their stereotypy may still limit the potential complexity of the observed neural signals [29]. Further, natural movements are rarely stereotyped or neatly divided into trials. The random target task dataset [44] also contains motor cortical data, but introduces different modeling challenges. It contains continuous, point-to-point reaches that start and end in a variety of locations, do not include delay periods, and have highly variable lengths, with few (if any) repetitions. These attributes preclude simple trial-averaging approaches to de-noise observed spiking activity. Further, we evaluate models using random snippets of the continuous data stream. The unpredictability of random snippets (*e.g.*, a new target could appear at any point in a data window) means that the simplification of autonomous dynamics is a poor approximation (discussed in [32, 8]).

`MC_RTT` spans 15 minutes of continuous reaching, artificially split into 1351 600ms "trials", and includes 130 neurons and simultaneously monitored hand kinematics. A number of other RTT datasets are available on Zenodo [44]. Successful modeling under this benchmark is predicated upon the ability to infer latent representations from single-trial data and infer the presence of unpredictable inputs to the population's activity.

**Area2_Bump.** Somatosensory areas may have substantially different dynamics from motor areas [45], owing to their distinct role of processing sensory feedback, which is critical to our ability to make coordinated movements. `Area2_Bump` includes neural recordings from area 2 of somatosensory cortex [46], an area that receives and processes proprioceptive information, or information about the movement of the body. These data were recorded as a monkey performed a simple visually-guided reaching task, where each trial consisted of a reach to a visually presented target using a manipulandum. However, in a random 50% of trials, the manipulandum unexpectedly bumped the monkey's arm in a random direction before the reach cue, necessitating a corrective response [46].

`Area2_Bump` includes 462 total trials and 65 neurons, and associated hand kinematics and perturbation information. Successful models would likely need to infer inputs to help describe activity after sensory perturbations, and should also be robust to low neuron counts.

**DMFC_RSG.** Dorso-medial frontal cortex (DMFC) is a high-level cognitive region, which poses unique challenges for LVMs. Neurons in high-level areas show mixed selectivity to different sensory stimuli and movement parameters [47]. Furthermore, behavior and task variables in cognitive tasks are usually not directly observable, which makes the task of inferring internal states even more critical. Additionally, the input and output layers of DMFC are not as clearly delineated as the other sensory or motor cortices [48]. `DMFC_RSG` contains recordings from DMFC while a monkey performed a "Ready-Set-Go" time-interval reproduction task [49]. In the task, animals estimated a time interval between two visual cues and then generated a matching interval by moving their eyes or hands toward the target cue. Uniquely, that time interval is itself a variable that animals had to infer based on sensory information, and then reproduce based on their internal time estimates. The multiple response modalities, target locations, timing intervals, and timing priors resulted in a total of 40 task conditions [49].

`DMFC_RSG` includes 1289 trials and 54 neurons, and associated task timing, condition, and reaction time information. High-performing LVMs would likely need to balance input- and internally-driven dynamics, and again be robust to low neuron counts.

## 3   NLB '21: Pipeline and Metrics

In addition to the diverse neural datasets that we provide as part of this work, we introduce an evaluation framework for evaluating LVMs across a number of axes, which may help the community assess how different design choices affect a model's relevance to the variety of potential challenges in neuroscientific applications, and help users from various areas of systems neuroscience determine which approaches are most relevant to their application.

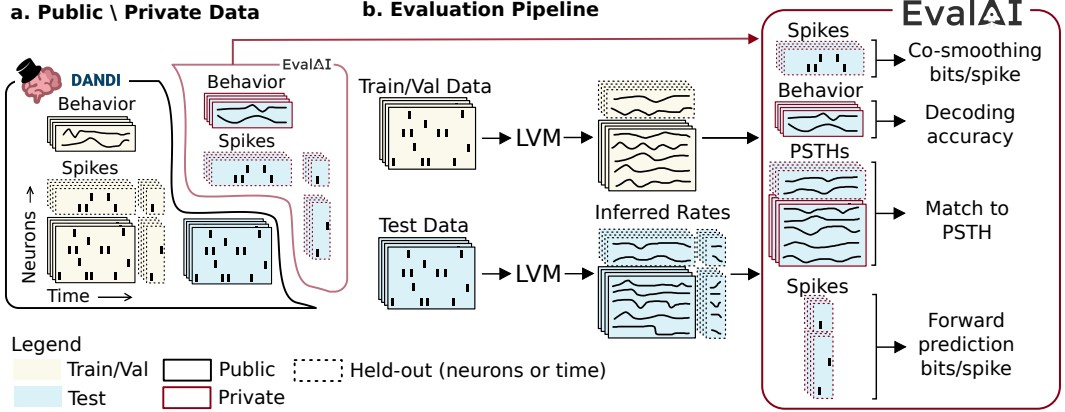

Figure 2: **Evaluation pipeline. a**. Datasets are split into several components to enable rigorous evaluation of the candidate model (described in Section 3.1). **b**. During evaluation, users submit firing rate predictions that are evaluated against private data (held on EvalAI servers) via co-smoothing (primary metric, Section 3.2), as well as match to PSTH, forward prediction, and behavioral decoding (secondary metrics, Section 3.3).

As motivated in the introduction, LVMs characterize coordination in a neural population's activity across space (*i.e.*, neurons) and time, such that high-performing models should be able to predict activity of neuron-time points that they do not have direct access to. Our primary evaluation metric thus measures an LVM's ability to predict the activity of held-out neurons (co-smoothing, described in Section 3.2). We also use a secondary metric that measures an LVM's ability to predict held-out timepoints (forward prediction, described in Section 3.3), and additional secondary metrics that are dataset-specific (behavioral prediction and PSTH-matching, described in Section 3.3).

## 3.1 Evaluation strategy and pipeline

To support evaluation, in addition to defining train/val/test splits for our datasets, we further designate neurons and timepoints as "held-in" or "held-out." While both held-in and held-out data, as well as behavioral data, are available for train/val trials, only held-in data are provided for test trials. Withheld test data are stored exclusively on EvalAI servers (Figure 2).

At evaluation time, users submit inferred firing rates for both train/val and test splits. Train/val inferred rates are only used to compute a simple linear decoder for behavioral prediction - held-out (future) timepoints are unneeded. Test inferred rates are evaluated via co-smoothing (primary metric), as well as match to PSTH, forward prediction, and behavioral decoding using the linear decoder (secondary metrics). Model predictions are submitted on EvalAI in 5 ms bins.

By requiring only that users submit predictions for the specific neuron-time firing rates, we do not place specific restrictions on the method by which candidate LVMs are trained or perform inference. For example, methods may leverage the provided behavioral data as additional information to guide learning of their latent representations, as done in [50]. Note also that we specify a val split as a standard for reporting ablations and analyses, though users are free to validate their model with other splits.

## 3.2 Primary evaluation metric: Co-smoothing

Our primary evaluation metric is the log-likelihood of held-out neurons' activity. As mentioned previously, the test data is split into held-in and held-out neurons (Figure 2a). Given the training data and the held-in neurons in the test data, the user submits the predicted firing rates $\boldsymbol{\lambda}$ of the held-out neural activity $\hat{\boldsymbol{y}}$. We use a Poisson observation model in the log-likelihood such that the probability of the held-out spike count of neuron $n$ at time point $t$ is $p(\hat{\boldsymbol{y}}_{n,t}) = \text{Poisson}(\hat{\boldsymbol{y}}_{n,t}; \boldsymbol{\lambda}_{n,t})$. The overall log-likelihood $\mathcal{L}(\boldsymbol{\lambda}; \hat{\boldsymbol{y}})$ is the sum of the log-likelihoods over all held-out neurons $n$ and time points $t$.

To normalize the log-likelihood score, we convert it to bits per spike using the mean firing rates of each held-out neuron [51]. Bits per spike is computed as follows:

$$\text{bits/spike} = \frac{1}{n_{sp} \log 2}(\mathcal{L}(\boldsymbol{\lambda}; \hat{\boldsymbol{y}}_{n,t}) - \mathcal{L}(\bar{\boldsymbol{\lambda}}_{n,:}; \hat{\boldsymbol{y}}_{n,t}))$$

where $\bar{\boldsymbol{\lambda}}_{n,:}$ is the mean firing rate for the neuron $n$ and $n_{sp}$ is its total number of spikes. Thus, a positive bits per spike (bps) value indicates that the model infers a neuron's time-varying activity better than a flat mean firing rate.

The approach of predicting the activity of held-out neurons conditioned on the held-in neurons on test data is referred to as co-smoothing [9] and provides a generalization of leave-neuron-out accuracy measures now commonly used in the neuroscience community [9, 12, 32, 52, 15]. Co-smoothing assumes that the latent variables underlying the activity of held-out neurons can be inferred from the training neurons supplied at test time. While this may not hold for all neurons in a population, the observation that latent representations are distributed across many neurons provides strong support for this assumption (reviewed in [12, 3]).

### 3.3 Secondary metrics

**Behavioral decoding.** Relating neural activity to behavior is a common goal in modeling neural data, and neural latent variables are often interpreted by identifying relationships between the latent states and behavioral variables of interest. Therefore, we included behavioral decoding accuracy given the inferred latent variables as a secondary benchmark metric.

For the `MC_Maze` , `MC_RTT` , and `Area2_Bump` datasets, behavioral decoding is evaluated by fitting a ridge regression model from training rates to behavioral data and measuring R2 of predictions from test rates. Though better decoding performance can be achieved with more complex decoders, we choose to enforce a linear mapping for all models to prevent excessively complex decoders from compensating for poor latent variable estimation. With all three datasets, the behavioral data used is monkey hand velocity; many kinematic variables are known to correlate with sensorimotor activity, but some (such as position) change slowly, making it easy to saturate decoding performance by heavy smoothing of the neural activity. We hypothesized that hand velocity would provide a challenging enough decoding target to differentiate the quality of the representations inferred by different models.

For `DMFC_RSG` , behavioral decoding is more difficult to evaluate (as described earlier). However, previous work has indicated that the rate at which the neural population state changes, or the neural speed, correlates negatively with $t_p$, the time between the Set cue and the monkey's Go response in the Ready-Set-Go (RSG) task [49, 8]. Thus, we compute average neural speed for each trial from test rate predictions, and then calculate Pearson's *r* between neural speeds and the measured $t_p$, as a measure of how well predicted neural activity reflects observed behavior [49].

**Match to PSTH.** When behavioral tasks include distinct conditions with repeated trials, a coarse, commonly-used method to de-noise spiking activity is to compute a peri-stimulus time histogram (PSTH). PSTHs are computed by averaging neuronal responses across trials within a given condition, and thus reveal features that are consistent across repeated observations of the behavioral condition [3]. For LVMs, inferring firing rates that recapitulate the PSTHs provides evidence that a model can capture certain stereotyped features of the neurons' responses. However, we use the match to PSTH as a secondary metric because of two limitations: First, not all datasets are well-suited to computing PSTHs. And second, the PSTH treats all across-trial variability as "noise", and thus matching the PSTH does not test a model's ability to predict single-trial variability that may be prominent and a key part of a given brain area's computational role [3].

Outside of the `MC_RTT` dataset, which is not well-suited to trial-averaging approaches due to its lack of clear trial structure, typical neural responses to specific conditions for the other datasets can be estimated by averaging smoothed spikes across trials within the same condition. We evaluate how well predicted rates match PSTHs by computing $R^2$ between trial-averaged model rate predictions for each condition and the true PSTHs. $R^2$ is first computed for each neuron across all conditions and then averaged across neurons.

**Forward prediction.** We additionally test the model's ability to predict the responses of *all neurons* at unseen, future time points. The forward prediction benchmark is evaluated in a similar manner as co-smoothing, with the distinction that the held-out responses are across all neurons at future time points. Forward prediction provides a further measure of how well a model can capture temporal structure in the data. However, it assumes that future neural activity can be predicted based on prior neural activity. This should not be the case in general, and is especially problematic for brain regions and behavioral tasks in which unpredictable inputs are common. Thus while forward-prediction provides some assessment of an LVM's ability to predict activity that is itself predictable, it is best applied to scenarios where data can be well-modeled via autonomous dynamics (such as `MC_Maze`).

| | Co-smoothing bps ($\uparrow$) | Behavior decoding ($\uparrow$) | PSTH $R^2$ ($\uparrow$) | Forward pred bps ($\uparrow$) |
|---|---|---|---|---|
| Smoothing | 0.211 | 0.624 | 0.183 | − |
| GPFA | $0.187_{\pm 0.001}$ | $0.640_{\pm 0.001}$ | $0.518_{\pm 0.002}$ | − |
| SLDS | $0.219_{\pm 0.006}$ | $0.775_{\pm 0.006}$ | $0.482_{\pm 0.036}$ | $-1.020_{\pm 0.943}$ |
| NDT | $0.329_{\pm 0.005}$ | $0.897_{\pm 0.009}$ | $0.579_{\pm 0.023}$ | $0.209_{\pm 0.010}$ |
| AutoLFADS | $0.346_{\pm 0.005}$ | $0.907_{\pm 0.002}$ | $0.631_{\pm 0.022}$ | $0.239_{\pm 0.003}$ |

Table 1: `MC_Maze` **metrics**. All NLB'21 metrics $\pm$ standard error of the mean applied to `MC_Maze`. Baseline rank order is consistent across metrics; deep neural networks (NDT and AutoLFADS) improve on less expressive baselines. High decoding performance across the board directly links between recorded activity and external behavior.

### 3.4 Baselines to seed the benchmark

We seed the benchmark with 5 established methods for modeling neural population activity: Smoothed spikes [12], Gaussian Process Factor Analysis (GPFA) [12], Switching Linear Dynamical System (SLDS) [53–55], AutoLFADS [8], and the Neural Data Transformer (NDT) [42].

These models vary in terms of their underlying assumptions on dynamics, the forward (or generative) model, and their overall complexity [2]. Spike smoothing is the simplest yet still common approach for de-noising firing rates by convolving spiking activity with a Gaussian kernel. GPFA models neural state as a low-dimensional collection of Gaussian processes and thus imposes a simple linear dynamics on the latent space of the generative model. SLDS expands upon the standard linear dynamical system and approximates complex non-linear dynamics by alternating or "switching" between multiple distinct linear systems. AutoLFADS is a variational sequential autoencoder that models neural dynamics with an RNN and thus can capture complex nonlinear dynamics and embeddings. NDT uses a transformer to generate neural activities, without any explicit dynamics model.

## 4 Results

### 4.1 `MC_Maze` as Neuroscience MNIST

We first provide a close examination of model results on `MC_Maze`, which has many desirable qualities for a basic model litmus test. It has rich structure with 108 different reaching conditions, 180 neurons, and $> 2000$ trials. Yet each condition also has many repeated trials, allowing assessment through PSTH metrics. Additionally, because motor cortex has close anatomical ties to motor output, `MC_Maze` supports model evaluation via behavioral decoding. Finally, it is well-established both empirically and theoretically, as many LVMs have already been evaluated on the dataset and the dynamics are well-approximated as autonomous [32], inviting forward prediction measures to assess the quality of an LVM's dynamics model. This well-controlled setting makes `MC_Maze` like a "Neuroscience MNIST": immediately useful for validation of neural LVMs, even if potentially limited for long term neuroscientific or modeling advances. The evaluation of our baseline models on this dataset is presented in Fig. 3 and Table 1.

For the baseline models, the rank ordering of performance is largely consistent across metrics, and deep networks (AutoLFADS, NDT) are stronger across the board. The consistency of rankings in `MC_Maze` demonstrates the validity of co-smoothing in a well-structured dataset, motivating its use in more challenging ones.

Note that we do not expect primary and secondary metrics to be perfectly correlated. For example, post-processing the AutoLFADS-inferred rates via smoothing decreases co-smoothing performance, but increases behavioral decoding performance ( Fig. 3e). This highlights a conflict between unsupervised (co-smoothing) and supervised (behavioral decoding) evaluations. Similarly, we observe that GPFA underperforms spike smoothing when assessed via co-smoothing, but outperforms it in behavioral decoding; following Fig. 3e, this may be because GPFA has inferred rates with coarse features that evolve even slower than the spike smoothing kernel, which is consistent with the single-trial inferences in Fig. 3c. Note also that further tuning of hyperparameters or the kernel parameterization could improve GPFA, and more advanced versions that are designed for spike count data [56] may achieve higher performance.

We also separately examine forward prediction performance; they are omitted for spike smoothing and GPFA which do not have clear adaptations for forward dynamics. Absolute performance of likelihood

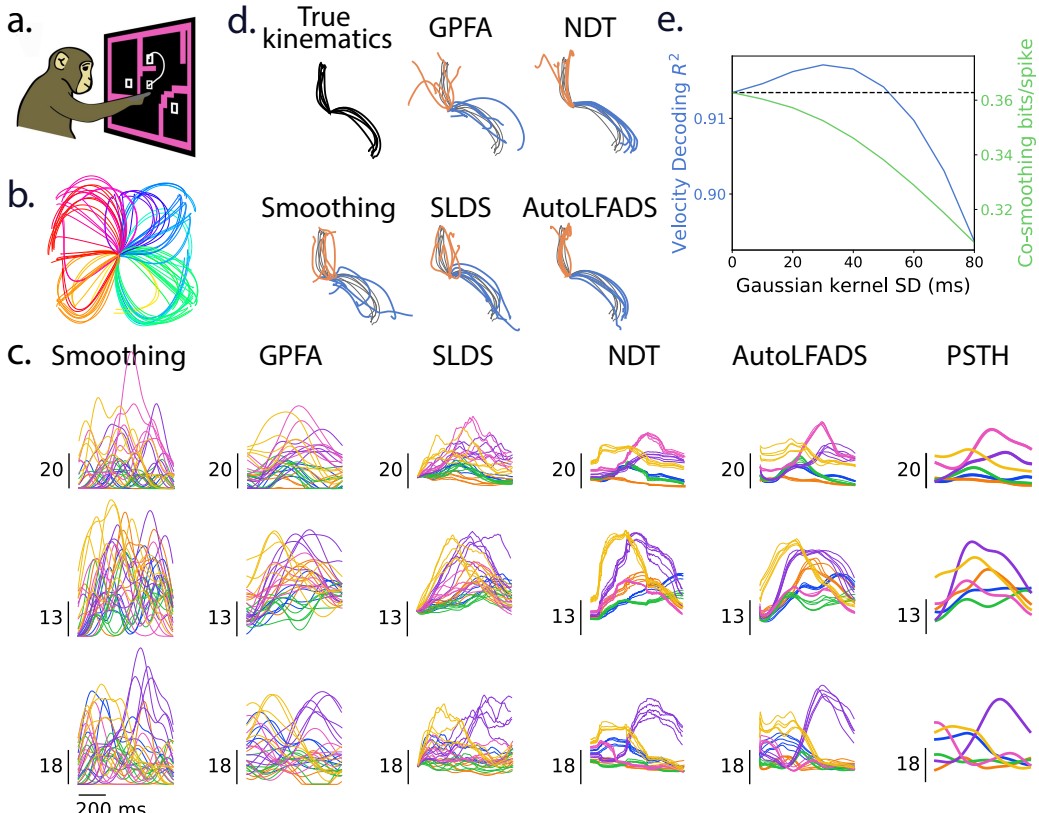

Figure 3: **MC_Maze in detail**. **a**. A monkey made curved reaches through a virtual maze. **b**. The task contained 108 different reaching conditions (colored by angle to target). **c**. (columns 1-5) Single trial inferred rates from the baseline models, and (column 6) PSTHs, for 3 neurons and 6 conditions (out of 108). Vertical scale shows spikes/second. Higher-performing models infer single-trial rates with greater across-trial consistency, that also resemble the trial-average. **d**. Single-trial hand trajectories are estimated via linear decoding of hand velocities based on inferred rates, and overlaid on the measured hand trajectories (grey). Shown are 5 trials each for 2 conditions. **e**. Behavioral decoding performance can be improved by smoothing the rates inferred by AutoLFADS, though co-smoothing performance decreases.

|  | MC_Maze | MC_Maze-L | MC_Maze-M | MC_Maze-S | MC_RTT | Area2_Bump | DMFC_RSG |
|---|---|---|---|---|---|---|---|
| Smoothing | 0.211 | 0.224 | 0.167 | 0.191 | 0.147 | 0.154 | 0.120 |
| GPFA | $0.187_{\pm 0.001}$ | $0.239_{\pm 0.001}$ | $0.172_{\pm 0.001}$ | $0.217_{\pm 0.002}$ | $0.155_{\pm 0.000}$ | $0.168_{\pm 0.000}$ | $0.118_{\pm 0.000}$ |
| SLDS | $0.219_{\pm 0.006}$ | $0.290_{\pm 0.008}$ | $0.210_{\pm 0.008}$ | $0.250_{\pm 0.001}$ | $0.165_{\pm 0.004}$ | $0.187_{\pm 0.005}$ | $0.120_{\pm 0.001}$ |
| NDT | $0.329_{\pm 0.005}$ | $0.362_{\pm 0.006}$ | $0.259_{\pm 0.021}$ | $0.251_{\pm 0.014}$ | $0.160_{\pm 0.009}$ | $0.267_{\pm 0.003}$ | $0.162_{\pm 0.009}$ |
| AutoLFADS | $0.346_{\pm 0.005}$ | $0.374_{\pm 0.005}$ | $0.304_{\pm 0.006}$ | $0.301_{\pm 0.009}$ | $0.192_{\pm 0.003}$ | $0.259_{\pm 0.001}$ | $0.181_{\pm 0.001}$ |

Table 2: **Co-smoothing across datasets**. We report co-smoothing bits/spike $\pm$ standard error of the mean across the datasets in NLB. While exact rankings and performance gaps vary per dataset, more expressive deep neural networks tend to perform the best. Note that absolute co-smoothing performance is not easily compared across datasets.

is generally uninterpretable, but SLDS scores are negative, implying poor fitting that underperforms a simple estimate that uses the average firing rates. We expect that the recurrent version of SLDS, an rSLDS, would improve on forward prediction performance. NDT and AutoLFADS both perform well beyond this null hypothesis, and comparing their forward prediction relative to co-smoothing, we conclude they are inferring the predictable, autonomous dynamics we expect in MC_Maze .

## 4.2 Results across datasets

The generality of co-smoothing allows its application across the diverse data from different brain regions (Table 2). The new datasets preserve overall rankings from MC_Maze , but provide more nuance. In contrast to the MC_Maze and Area2_Bump , which both show a clear gap in results

between linear and deep models, we observe a smaller gap in `MC_RTT`. `DMFC_RSG` in particular demonstrates the need for deep, expressive LVMs in identifying structure in deep cognitive areas, since only deep networks identify structure better than the simple spike smoothing baseline.

The different datasets have varied numbers of neurons with variable firing rates. As such, direct comparisons of scores across datasets is tricky. However, by using multiple datasets from the same underlying experiment (as in `MC_Maze-{S,M,L}`), we can make more direct comparisons and also study the scaling of different methods across different data sizes. We find that both the NDT and AutoLFADS scale well across small sample sizes, with AutoLFADS providing more stable performance on smaller datasets and NDT lagging slightly for `MC_Maze-S`.

## 5 Discussion

NLB'21 is a substantial initial step towards standardized evaluation of LVMs on neural data. We hope that in highlighting the ability of LVMs to extract population structure across the brain, we can catalyze the broad adoption of LVMs in systems neuroscience.

Our model-agnostic framework enables standardized comparison of 5 approaches that vary in complexity and assumptions about latent structure, across 7 different real neuroscientific datasets. This baseline evaluation is already, to our knowledge, the broadest LVM comparison yet attempted, and the platform will allow the comparison to be greatly extended via community-driven contributions.

Our initial evaluation delineates deep networks as powerful tools for uncovering structure from neural population activity. We anticipate the benchmark as a conduit for ML researchers with cutting-edge approaches to evaluate those approaches on neuroscientific datasets. To be inclusive to these methods, we do not preclude alternate model training strategies (*e.g.* data augmentation, transfer learning, etc.), nor do we restrict by computational or memory requirements, though we do encourage reporting these details on submission.

**Extensions and Limitations.** NLB'21 is our first suite in a planned effort to rigorously evaluate models of neural data. We anticipate extensions, for example, towards modeling multi-region interactions, alternate recording modalities, transfer learning, and robust state estimation under recording instability. A particularly important direction for future work, and a subject of interest for the broader ML community, is developing measures for evaluating model or latent state interpretability, as interpretability is key to an LVM's ability to drive scientific progress.

**Ethical considerations.** All datasets were collected under procedures and experiments that were approved by the Institutional Animal Care and Use Committees at the respective institutions. Specifics of the experimental procedures can be found in the primary references for each dataset. Animal models are a cornerstone of research to increase our understanding of the brain, and we hope that benchmarking efforts that improve standardization, such as NLB, might help minimize redundancy in data collection, while also enabling the development of LVMs that allow researchers to get more use from existing data.

Separately, advances in LVMs might be used to improve the performance of brain-machine interfaces or other therapeutic devices related to brain injury or disease [4]. While such devices are largely targeted towards restoring function to people with disabilities or impairments, it is important to consider the potential impacts on fundamental aspects of the human experience, such as identity, normality, authority, responsibility, privacy, and justice, which are the subject of ongoing study [57, 58].

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
