# OpenReview forum: "Neural Latents Benchmark ‘21: Evaluating latent variable models of neural population activity"
_NeurIPS.cc/2021/Track/Datasets_and_Benchmarks/Round2 — NeurIPS 2021 Datasets and Benchmarks Track (Round 2)_

### Official Review · Reviewer_iRhp · 2021-09-14
**Review of the Neural Latents Benchmark '21**

**Rating:** 7
**Confidence:** 3

**Strengths:**

This work is quite significant for improved benchmarking of different algorithms that model neural activity data. There has been an increased interest in understanding of neural activity through latent variable models for both forward prediction as well as behavior prediction. Given this increased interest, a proper benchmark with diverse task candidates and evaluation metrics is quite important for future research in this area.

The existence of private data for providing a leaderboard and preventing overfitting by the research community is another positive trait of the provided benchmark.

The consistency of different evaluation metrics on the models considered provides confidence about the validity of the data and the metrics.

**Weaknesses:**

While the authors evaluate multiple relevant models on the data that they curate, they do not provide standard deviation corresponding to all the metrics. It would be nice to perform evaluation over multiple random seeds and provide standard deviation of all the metrics presented in the paper.

Throughout the start, the paper claims that progress in latent variable models is impeded by lack of standardized datasets and evaluation metrics. This feels like an over-claim because LVMs typically extend beyond neural activity data and there are a lot of standardized datasets and evaluation metrics that people have developed for evaluation of LVMs in general settings. It would be nice to establish early on that the paper concentrates on standardization of LVM evaluation on **neural activity data**.

**Additional Feedback:**

It would be nice to see the discrete latents associated with SLDS on the provided data to see if some discrete interpretable latents are being observed on it.

**Clarity:**

The paper is clearly written and is generally easy to follow through. The organization of evaluation metrics, baselines and codebase pipeline could be better.

**Correctness:**

The dataset is constructed in a sound way and the evaluation methods and experiment design are appropriate for the tasks introduced in the paper. The authors discuss in detail how each task that they provide is curated in the Appendix.

**Documentation:**

The paper provides sufficient details for how the data has been obtained and provides a well functioning webpage detailing further details about it. The authors further open-source their code for reproducibility and provide detailed comments about different model implementations and methodology for curating the data in the Appendix.

**Ethics:**

There are no ethical concerns that I am aware of.

**Relation To Prior Work:**

The authors do a good job at discussing existing work in the field and identifying the need for standardization of evaluation. They also do a pretty good job at reporting the performance of a wide variety of models ranging from more early approaches like Gaussian smoothing to Deep Neural Network based approaches.

**Summary And Contributions:**

This work introduces a benchmark for evaluation of latent variable models for neural population activity data. The authors conduct experiments on monkey by making it do a variety of different tasks and recording both its behavior and neural activity. The paper does a thorough job at evaluating different types of algorithms on the curated data and also use different evaluation criteria. Given the variety of tasks across which data is obtained and the existence of private data for model ranking to prevent overfitting to test data, I believe that the paper provides important contributions to the field.

---

> ### Author Response · Authors · 2021-09-30
> **Response to iRhp**
>
> We thank the reviewer for their in depth comments and enthusiasm for the work.
>
>
> > 1. It would be nice to perform evaluation over multiple random seeds and provide standard deviation of all the metrics presented in the paper.
>
>
> We appreciate this point and agree the lack of statistical comparisons was a shortcoming of the original submission. (Rev. Ty7Z made a similar comment). We addressed this in two ways: 1) Per Rev. Ty7Z’s suggestion, we performed a bootstrap analysis in which we compared each pair of baselines on 50 bootstrapped samples of the test trials. These results are presented in Appendix A.1.1.2) Per the reviewer’s suggestion, we re-ran each method with different random seeds to assess variability across model runs. Our main results (Tables 1 & 2) now reflect the S.E.M. found across seeds.
>
> These new analyses confirmed that the original model rankings and conclusions are statistically sound and robust to random seeds and the specific test split.
>
>
> > 2. It would be nice to establish early on that the paper concentrates on standardization of LVM evaluation on neural activity data.
>
> Thank you for the comment - indeed we missed the instances early on where we had not specified that the LVMs we are evaluating are applied to neural activity. We have corrected this phrasing in the abstract and introduction.
>
>
> > 3. It would be nice to see the discrete latents associated with SLDS on the provided data to see if some discrete interpretable latents are being observed on it.
>
> This is an astute observation, i.e., that the switching latent dynamics uncovered by SLDS models can be used to interpret neuronal population activity. However, interpretability of SLDS latent dynamics is a bit outside the scope of this benchmark, and is better explained in the SLDS papers that we reference, e.g. [1,2].
>
> [1] Dynamical segmentation of single trials from population neural data. Petreska et al. NeurIPS, 2011.
> [2] Tree-Structured Recurrent Switching Linear Dynamical Systems for Multi-Scale Modeling. Nassar et al. ICLR, 2019.

---

> > ### Comment · Reviewer_iRhp · 2021-10-03
> > **Reponse to Authors**
> >
> > Thank you for the response and for adding the bootstrapped and variability experiments/analysis. This essentially addresses most of the concerns I had regarding the work.

---

### Official Review · Reviewer_bqKG · 2021-09-20
**New benchmark for evaluating LVMs for neural activities**

**Rating:** 5
**Confidence:** 2
**Correctness:** See above.
**Clarity:** The paper well-written and easy to fo…

**Strengths:**

1. A standardized evaluation suite for latent variable modeling on neural activity data.
2. Propose new evaluation metrics for LVMs.


**Weaknesses:**

I think the proposed "behavior decoding accuracy" is a very important metric. Because the proposed benchmark is specifically for LVMs so it would be important to directly evaluate the quality of the latent representations. However, I have some concerns about this metric:
1. To my understanding, the behavioral data (monkey hand velocity) is also provided in the train/eval data. Are the LVMs supposed to use this data to guide the learning of the latent representation? If this is the case, then the latent representation could be overfitting to the behavioral data.
2. One specific kind of behavior, monkey hand velocity, is used for behavior decoding. This could introduce a bias toward this specific kind of behavior. Also according to Table 1, the AutoLFADS can already get a more than 0.9 R2 score, which leaves little space for other models to improve.

I also have some concerns regarding the related work, I'll elaborate below in the Relation To Prior Work section.

**Additional Feedback:**

See above.

**Documentation:**

Documentation looks good to me.

**Ethics:**

I don't have any ethical concerns.

**Relation To Prior Work:**

I suspect that there are other neural activity datasets (not necessarily benchmarks) that are used previously in evaluating LVMs. However, I didn't find any related works discussed along this line. Please correct me if I missed something or the authors should give a comprehensive discussion on this. I think it is important to make the difference between this dataset and the existing datasets clear, or it will be hard to evaluate the contribution of this paper.

On the other hand, latent variable models usually refer to models that infer/learn latent variables from the observed variables. Not all unsupervised learning models or generative models are LVMs. Specifically, I don't think MLMs like BERT are LVMs. And the evaluation of MLMs does not correspond to the evaluation of "latent representations". I think the authors should do a more thorough literature review on the evaluation methods of LVMs in machine learning researches. One common evaluation method for LVMs is to interpolate the latent space along one direction and observe the meaningful transition in outputs. This does not necessarily correspond to a provided label or category information.

**Summary And Contributions:**

This paper proposes a standard benchmark for evaluating latent variable modeling (LVM) of neural population activity. Four diverse datasets are proposed along with new metrics and baseline models.

---

> ### Author Response · Authors · 2021-09-30
> **Response to bqKG (1/2)**
>
> We thank the reviewer for their thoughtful response, and appreciate their compliments on the paper’s clarity.
>
>
> > 1. “Behavior decoding accuracy is an important metric”
>
>
> We agree that behavior decoding has its role as a direct evaluation of latents and provides context for model performance for certain use cases (such as for brain-computer interface decoding of movement intent). This is why we include behavioral labels that were measured during neural data collection and use behavior decoding as a secondary metric. However, as described in Section 1.1, since we do not know what behaviors the neural latents in any given area should encode, behavior decoding cannot serve as a generalizable metric; in contrast, our primary metric (co-smoothing) provides a general purpose, brain area- and task-agnostic metric to evaluate latent variable models of neural activity. The co-smoothing evaluation strategy is explicitly testing a model’s ability to predict held-out neural activity, i.e., responses of some neurons that it does not have access to at test time. On the test examples, only a fraction of the neural responses are supplied to the model, and those supplied responses are used to generate predictions for the held-out neurons on these trials. To draw an analogy to computer vision, we are essentially masking a portion of pixels in an image and asking whether a model can predict those unseen pixels at test time given the rest of the pixels. This strategy is schematized in Fig. 2 and described in detail in Section 3.2.
>
> By virtue of being unsupervised, the co-smoothing approach avoids the pitfall of “bias towards [a] specific kind of behavior” as mentioned by the reviewer.
>
> > 2. Are the LVMs supposed to use this data to guide the learning of the latent representation?
>
> This is an excellent question. LVMs are quite often fully unsupervised, i.e., many approaches do not use behavioral information to guide learning of the latents. However, some emerging methods do (e.g. [1]). We thus made our benchmark agnostic to this issue - models may use behavioral information to guide learning of the latents, and all metrics can be evaluated regardless of this design choice. This gives modelers maximum flexibility. Of course, as the reviewer astutely notes, models that explicitly take behavior into account when learning latent representations could risk biasing their latent estimates towards that behavior. We appreciate the observation, and now clarify this in Section 3.1.
>
> [1] Modeling behaviorally relevant neural dynamics enabled by preferential subspace identification. Sani et al. Nature Neuroscience, 2021.
>
>
> > 3. Little space for models to improve in behavioral decoding on Maze.
>
> The reviewer points out that on the Maze dataset, there is little room for improvement in decoding accuracy. However, we note that even when a model achieves a high level of decoding accuracy, there can still be a lot of room for improvement in generating high-quality predictions of held out neurons. To clarify this point, we ran new analyses where we behavior decoding is compared with the co-smoothing metric (A.1.1). We find that a wide range of models can achieve good behavior decoding but vary in the quality of their predictions on held-out neurons. This underscores the importance of using an unsupervised measure of the recovery of firing rates, and going beyond behavior.
>
> > 4. Other datasets previously used in evaluating LVMs, difference between these datasets and previous datasets.
>
> We thank the reviewer for pointing out this important missing context in our related work. We have lengthened our related work (Sec 1.2) and added a supplementary table (A.2) with commentary on previous neural data LVM datasets. There are few well-established datasets that many research groups have reported on, likely since access to these datasets tends to be restricted to those with personal connections to the experimentalists who collected the data. The datasets we use are not different in spirit from previously-collected datasets; however establishing this benchmark - with its accessibility, fixed pre-processing, standardized evaluation, lowered barrier to entry, and dedicated organizational efforts to encourage adoption - is critical in accelerating progress in the LVM space.

---

> > ### Author Response · Authors · 2021-09-30
> > **Response to bqKG (2/2)**
> >
> > > 5. I think the authors should do a more thorough literature review on the evaluation methods of LVMs in machine learning researches. One common evaluation method for LVMs is to interpolate the latent space along one direction and observe the meaningful transition in outputs.
> >
> > We agree that in ML more broadly, there are many tools for evaluating latent spaces. However, these tools make assumptions for what constitutes a desirable latent space. When evaluating LVMs in neuroscientific applications, it is unclear what constitutes a desirable latent space, and this might vary depending on the specific brain area under study, or the behavioral task being performed. These unknowns prevent supervised decoding strategies as well as techniques like latent space interpolation for general purpose evaluation. However, a general property of a good latent space is that it is informative about the observed variables - namely, in this case, the neuronal responses. Thus as our primary metric we choose co-smoothing, which (as mentioned earlier) evaluates the ability of the latents to capture features of the observed neuronal responses, and thus provides a brain area- and task-agnostic evaluation of LVMs.
> >
> > > 6. Not all unsupervised learning models or generative models are LVMs. Specifically, I don't think MLMs like BERT are LVMs.
> >
> > This is a good point. We agree that the original text may have made it unclear that our benchmark is specifically intended for LVMs for neuroscience, which, to our knowledge, are all generative models. We added revisions to section 1.1 (first paragraph) and 1.2 (first paragraph) to clarify these points. And in particular, we hope that the new supplementary table (A.2) makes clear the class of models that the benchmark was intended to evaluate.
> >
> > However, while we designed our evaluation strategy with neural data LVMs in mind, the benchmark can also evaluate models that might not be traditionally considered LVMs. For example, the NDT model tested in this work is based on the BERT MLM model. By aiming for maximum flexibility in model evaluation, an ancillary benefit is that the benchmark is compatible with evaluating models that are not LVMs.

---

> > > ### Comment · Reviewer_bqKG · 2021-10-06
> > > **Thank you for the response**
> > >
> > > I thank the authors for putting effort to answer my concerns. However, I'm not quite convinced and I'm inclined to keep my score unchanged. The reasons are elaborated below:
> > >
> > > First, about the definition of LVMs, I don't think there exists a "neuroscience LVM" definition that is different from the usual LVM definition. As you mentioned Neural Data Transformers (NDT) as a "neuroscience LVM" example, I checked the referenced paper and do not find the authors refer to NDT as an LVM at all. Instead, they called NDT "an architecture for modeling neural population spiking activity". I believe the authors have mixed LVMs with self-supervised representation learning models. I suggest the authors replace the phrase "latent variable model" in their title.
> > >
> > > Second, about behavior decoding accuracy which is intended to evaluate the quality of the learned latent, I think it is important because the authors claim they try to evaluate LVMs. An important part of LVM evaluation is to evaluate the quality of the learned latent. However, this evaluation metric cannot truly reflect the latent variable quality as users can use the provided behavior data to guide the learning of latent variables to bias towards this metric. In this case, the learned latent cannot be fairly compared which is a big design problem.
> > >
> > > Although the proposed benchmark could be useful for the neuroscience community, I think there are some major mistakes/defects of this benchmark that should be fixed before publishing, so I decide to not change my score.

---

> > > > ### Author Response · Authors · 2021-10-07
> > > > **Some clarifications**
> > > >
> > > > 1. Definition of LVMs.
> > > >
> > > > We appreciate the reviewer’s response, but think perhaps they misinterpreted our comment. We were not calling the NDT an LVM. We noted the NDT as an example of a model that is /not/ an LVM (agreeing with the reviewer that BERT and similar models are not LVMs). Yet, the NDT can be evaluated within the framework we present here. (For reference, the authors of the NDT paper are co-first and senior authors of the current work.) We note in our response that we designed our benchmark with neural data LVMs in mind, hence the use of the term in the title. However, by aiming for maximum flexibility in model evaluation, an ancillary benefit is that the benchmark is compatible with evaluating models that are not LVMs.
> > > >
> > > > Also, our use of “neuroscience LVM” was meant to describe latent variable models that have been used to model neural data. We agree that there is not a separate definition of latent variable models for neuroscience.
> > > >
> > > >
> > > > 2. Behavior decoding metric.
> > > >
> > > > We want to clarify two potential points of confusion regarding the use of the behavioral decoding metric.
> > > >
> > > > First, in case unclear: the behavioral data is /not/ provided with the evaluation (test) dataset, so a modeling approach cannot overfit to the behavioral data at test-time evaluation or use the test-time behavioral data to inform predictions of held-out neural activity. The behavioral data is provided in the training set, as not doing so would exclude existing LVM approaches from being evaluated in the benchmark (e.g., [1,2]). The strategy regarding holding in/out behavioral data is shown in Fig. 2.
> > > >
> > > > Second, the primary evaluation metric - used for ranking models - is co-smoothing.  This metric does not use the behavioral data and instead evaluates models based on predictions of held-out neural activity. Thus our benchmark provides holistic model evaluation regardless of whether the model is fit using the behavioral variables in the training dataset. All metrics are reported for every model, and thus, should a given model bias its latents toward the provided behavioral data, that will be reflected in a decrease in co-smoothing score (the primary metric).
> > > >
> > > > —
> > > >
> > > > Would the reviewer be willing to re-read our response with these clarifications in mind? We put substantial effort into carefully considering the reviewer’s feedback and addressing it with updates to the manuscript.
> > > >
> > > > ----
> > > >
> > > > [1] Modeling behaviorally relevant neural dynamics enabled by preferential subspace identification. Sani et al., Nature Neuroscience, 2021. Jan;24(1):140-149. doi: https://doi.org/10.1038/s41593-020-00733-0
> > > >
> > > > [2] Where is all the nonlinearity: flexible nonlinear modeling of behaviorally relevant neural dynamics using recurrent neural networks. Sani et al., bioRxiv 2021. doi: https://doi.org/10.1101/2021.09.03.458628

---

### Official Review · Reviewer_Ty7Z · 2021-09-20
**Interesting benchmark resource for neuroscience**

**Rating:** 7
**Confidence:** 3
**Correctness:** ML evaluation could be improved as su…
**Clarity:** Yes

**Strengths:**

The authors do a good job introducing a set of benchmark data consisting of controlled neuronal experiments with rhesus monkeys. The data are described clearly and the benchmark is motivated well. The use of a hold-out test set and competition-style format make the resource even more attractive from a benchmarking perspective. The paper is well-written and the repository includes code for reproducing their baseline reporting. The data are also well-illustrated.

**Weaknesses:**

The biggest weakness is that the method evaluations lack any statistical comparisons or confidence estimates. As a result it is not possible to say whether any of these methods are significantly better than any other. The algorithms should be compared over multiple training instances with shuffled data, or a bootstrap analysis should be performed to estimate performance variability among the methods for each task.

- it seems that this data was collected in 2010, and has been the subject of several research papers. With that in mind, the authors should do more to explain why such a benchmark is necessary by contrasting with findings from prior work with these data.



**Additional Feedback:**

- "MC_Maze as a Neuroscience MNIST" is repeated several times. MNIST is important to the history of deep learning of course, but IMO these days it is seen as an over-used / unrealistic benchmark with rampant over-fitting to the test set.

**Documentation:**

Yes

**Ethics:**

The authors state they foresee no ethical issues or negative societal impacts of their study, yet I can't imagine that to be true. This data is collected from living beings (see Ethics guidelines #1) and I would think the process of implanting electrode arrays in rhesus monkeys, and keeping them as laboratory animals, inspires a lot of ethical discussion. Even if this is a study centered on basic science, I would like to see the authors address potential harm that can come to these animals as a result of their use. This line of research has many potential outcomes, from more invasive experimental studies to invasive ML technologies that arise from improving our understanding of neural processes in the brain.

**Relation To Prior Work:**

Prior work is cited, but as I mention above it would be good to do more to distinguish and motivate this benchmark effort from previous work with these data.

**Summary And Contributions:**

I have no background in neuroscience but I enjoyed learning about this benchmark resource for latent variable modeling of neural population activity.

---

> ### Author Response · Authors · 2021-09-30
> **Response to Ty7Z (1/2)**
>
> We thank the reviewer for their in depth comments and appreciate their enthusiasm for the work.
>
> > 1. Lack of statistical comparisons: “The algorithms should be compared over multiple training instances with shuffled data, or a bootstrap analysis should be performed to estimate performance variability among the methods for each task.”
>
> We appreciate this point and agree the lack of statistical comparisons was a shortcoming of the original submission (Rev. iRhP made a similar comment). We addressed this in two ways: 1) Per the reviewer’s suggestion, we performed a bootstrap analysis in which we compared each pair of baselines on 100 bootstrapped samples of the test trials. These results are presented in Appendix A.1.2. 2) As suggested by Rev. iRhP, we re-ran each method with different random seeds to assess variability across model runs. Our main results (Tables 1 & 2) now reflect the S.E.M. found across seeds.
>
> These new analyses confirmed that the original model rankings and conclusions are statistically sound and robust to random model training seeds and the specific test split.
>
> In the limited rebuttal period, we were unable to perform a data shuffling analysis (e.g., shuffling the train and test sets and assessing the consistency of conclusions). We believe the new analyses mentioned already address this point conclusively, but we would consider including data shuffling for a potential camera-ready submission if the reviewer feels strongly about it.
>
>
> > 2. “it seems that this data was collected in 2010, and has been the subject of several research papers.
>
> Thank you for this comment - we agree this may have been confusing in the original submission. These datasets have been used in previous studies, and the Maze dataset in particular (first published in 2010) has been used extensively. However, their primary use has been in testing specific neuroscientific hypotheses rather than LVM development (e.g., for understanding the structure of neural population activity during different phases of movement). Additionally, many of the other datasets provided are collected in more recent years and thus offer a wealth of possibilities in terms of new analyses that could be possible as LVMs advance.
>
> We have now clarified that none of these datasets have been used for benchmarking purposes in the first paragraph of Section 2. To address the specific use of the Maze data in previous papers, we modified the description of the Maze dataset in Section 2 to clarify that much of its previous use has been for neuroscientific hypothesis testing, with some additional applications to LVM development (second paragraph of the MC_Maze subsection).
>
> > 3. With that in mind, the authors should do more to explain why such a benchmark is necessary by contrasting with findings from prior work with these data”
>
> To help readers understand the fragmented landscape that motivated the creation of this benchmark, we created a supplementary table that chronicles LVMs for neuroscience that have been presented at ML conferences, noting the lack of standardization in datasets and metrics (Supp. Table 3, section A.2). We also created a new paragraph to open the related works section (1.2): “Neural data LVM evaluation strategies,” which describes the challenges of the previous landscape and diversity of datasets, and points readers to the new Supplementary Table.
>
>
> > 4. Ethical concerns.
>
> The reviewer’s concern is well-placed, and we agree a section on ethical considerations is appropriate. While a broad discussion on the ethics of laboratory animal research is beyond the scope of this work, it is important to note that all data was collected under the guidelines of the Institutional Animal Care and Use Committees (IACUC) at the respective institutions. We hope that benchmarking efforts that improve standardization, such as NLB, might help decrease laboratory animal use by minimizing redundancy in data collection, while also enabling the development of LVMs that allow researchers to get more use from existing data. Additionally, LVM development might affect therapeutic devices related to brain injury or disease, which as the reviewer notes, have several ethical implications. Pointing readers to references on the topic may increase awareness. We appreciate the reviewer’s comment and have now added an additional “Ethical considerations” section at the end of the Discussion to clarify these points.

---

> > ### Author Response · Authors · 2021-09-30
> > **Response to Ty7Z (2/2)**
> >
> >
> > > 5. MC_Maze as Neuroscience MNIST
> >
> > We indeed agree that DL may have outgrown MNIST, despite its foundational relevance. We hope to place MC_Maze in the same role, as an introductory, foundational benchmark for the field of computational neuroscience. To this end, we have clarified that MC_Maze’s relative simplicity may limit its long term utility in Sec 4.1 (where we make the comparison to MNIST). The reviewer’s concern that a single MNIST-like dataset may result in rampant overfitting is well-placed, and hence we have chosen to introduce MC_Maze among a diverse set of datasets. We look forward to a future where new recording technologies and behavioral paradigms yield even more versatile datasets that further push the boundaries of LVM capabilities.

---

### Official Review · Reviewer_epMK · 2021-09-22
**Neuronal Spiking Recording benchmarks**

**Rating:** 4
**Confidence:** 3
**Clarity:** The paper is not clear to the non-neu…

**Strengths:**

The main strength of this work is the focus on providing a benchmark for unsupervised learning. This is often harder to do, than associating or annotating data with a few label classes. The authors also provide and test evaluation metrics for latent variable models of this type of data including co-smoothing and behavioral decoding.

**Weaknesses:**

1. This paper is very difficult to read for the general machine learning researcher. it assumes a lot of prior knowledge in a narrow area of neuroscience. For instance the specific readings that the authors provide are very different than the more common fMRI datasets used in computation and the authors do not state basic assumptions about the data types. For example it is unclear what pre-processing steps have to be taken to go from raw data to spike information, and presumably it is a lot since there are whole benchmark suites designed for this.

2. The authors conflate machine learning terms. They call their LVMs "generative models," whereas they seem to merely be autoencoding or low-rank approximating the recording i.e., reconstructing the entire recording from lower dimensional latent variables.

3. The held-out data is termed strangely. This is just the standard test/training split in ML, the authors just find compressed representations of recordings I believe and test reconstruction rather than any sort of "prediction," as the authors say. The term co-smoothing is a bit strange to me as well and the origin of that is not mentioned.

4. Moreover, the authors don't seem to be seeking input from the ML community on what to do with latent representations, there is no mention of visualization, clustering, trajectory analysis or other tasks on the embedded space but only narrow references to tasks previously done in this subfield of neuroscience. This the variety of latent variables that they seek seem very limited to the types that are in references such as GLMs and gaussian processes.



**Additional Feedback:**

I would rewrite this submission for a more general audience.

**Correctness:**

The dataset seems to be sound, as do the experiments. However there are not enough experiments examining latent variables generated by modern machine learning methods.

**Documentation:**

The code and data seem well documented.

**Ethics:**

None.

**Relation To Prior Work:**

It is not clear to me what the relationship between Neural Latent Benchmarks and Neurodata without borders and DANDI is exactly. It seems these datasets are already available on DANDI . https://dandiarchive.org/dandiset/000070/

**Summary And Contributions:**

The authors provide datasets of recorded neuronal activity from populations of neurons in various brain regions. The intend for these benchmarks to facilitate latent variable models of neuronal recordings and propose a type of held-out validation called co-smoothing for evaluation.

---

> ### Author Response · Authors · 2021-09-30
> **Response to epMK (1/2)**
>
> We appreciate the reviewer’s feedback. As discussed in detail in the point-by-point responses below, the reviewer’s comments revealed ways in which readers with a more general ML background might misunderstand the presented material. We hope the additions and clarifications make our revised manuscript more accessible to the more general audience.
>
> > 1. Inaccessibility to the non-neuroscientist reader
>
> We thank the reviewer for their feedback on the clarity of the manuscript to non-neuroscience experts. We agree that the description of the pre-processing of datasets may have been inaccessible to the broader ML community. We have accordingly revised the second paragraph of Section 2 to provide an overview of data collection and pre-processing.
>
> > 2. “Narrow area of neuroscience”
>
> The reviewer stated that the benchmark focuses on a “narrow area of neuroscience”. We disagree - spiking recordings are of enormous import in neuroscience, and latent variable modeling for neural population spiking activity has established itself beyond neuroscience to play a growing part in ML research. However we recognize that this may have been unclear in the original submission. To provide context on the growth of this subfield in ML, we have added a supplementary table (Table 3, in A.2), which demonstrates the rich history and growing prevalence of LVMs for neural activity in ML/AI conferences. Neuroscience has always been a key area of interest at NeurIPS; however, as evident from the recent progress highlighted in Table 3 (A.2), development of LVMs of neural population activity is on the rise. Importantly, as shown in the table, previous efforts lacked standardization around datasets and metrics. This is the core motivation for the current benchmark: an attempt to facilitate comparisons, enable reproducibility, and lower the barrier to entry for ML experts to make impact.
>
>
> > 3. Potential conflation of machine learning terms (generative models and LVMs)
>
>
> We agree that the term generative model may have different meanings in different contexts. In our work, we use “generative model” to describe any model that can generate a neural activity pattern at its output. This would include frameworks like GANs or VAEs, as well as low-rank approximations that have an observation model for generating in the high-dimensional data space. All of these models can be used to generate outputs (neural spike trains) based upon a forward map from the model’s latent space, and an appropriate statistical model.
>
> It is possible that the term “latent variable model” might be similarly ill-defined. In our usage, LVMs are models that use unobserved (latent) variables to succinctly describe observed data. There is certainly overlap between LVMs and generative models, and to our knowledge, all LVMs for modeling neural population activity are generative models, in that they contain an observation model that maps from the latent variables to the observed high-dimensional space.
>
> Importantly, we agree with the reviewer that clarifying our terminology could be helpful for a broader ML audience. Accordingly, we revised the text in section 1.1 to more clearly define our usage of LVMs, and also highlight a recent review article on LVMs in neuroscience to guide readers with more general ML backgrounds. Additionally, the added Table 3 in A.2 provides specific examples of previous LVMs in neuroscience, to provide further context for our use of the term.
>
> We also note that, while we designed our evaluation strategy with LVMs in mind, the benchmark can also evaluate models that might not be traditionally considered LVMs. For example, the NDT model tested in this work does not contain a succinct description of the observed data. (This is discussed further in response to Reviewer bqKG.)

---

> > ### Author Response · Authors · 2021-09-30
> > **Response to epMK (2/2)**
> >
> >
> > > 4. The held-out data is termed strangely. This is just the standard test/training split in ML, the authors just find compressed representations of recordings I believe and test reconstruction rather than any sort of "prediction," as the authors say.
> >
> > We think this comment may reflect a misunderstanding of our evaluation strategy. While the reviewer argues that we do not test prediction, in fact the co-smoothing evaluation strategy is explicitly testing a model’s ability to predict held-out neural activity, i.e., responses of some neurons that it does not have access to at test time. On the test examples, only a fraction of the neural responses are supplied to the model, and those supplied responses are used to generate predictions for the held-out neurons on these trials. To draw an analogy to computer vision, we are essentially masking a portion of pixels in an image and asking whether a model can predict those unseen pixels at test time given the rest of the pixels. This strategy is schematized in Fig. 2 and described in detail in Section 3.2.
> >
> > After these clarifications, we hope that the reviewer can reconsider their score. If confusion still remains, we would be happy to clarify the evaluation strategy further.
> >
> >
> > > 5. The term co-smoothing is a bit strange to me as well and the origin of that is not mentioned.
> >
> > We point the reviewer to Section 3.2, which provided a detailed description of co-smoothing and referenced ML work that used the terminology when evaluating LVMs in neuroscience applications. To our knowledge, this term has been in use since at least Macke et al. 2011 [1]. We hope that our explanation above in (4) regarding our evaluation strategy helps to shed further light on how we use co-smoothing to evaluate models of neural activity.
> >
> >
> > > 6. Using techniques from the broader ML community to evaluate latents.
> >
> > While we agree that there is a wide range of work in different domains that focus on developing tools for evaluation of latent spaces, often through using “visualization, clustering, trajectory analysis”, all of these tools make assumptions on what constitutes a desirable latent space. As a result, such metrics cannot be applied across the wide range of different datasets provided in this benchmark. Instead, we decided to focus our efforts on establishing an evaluation metric that can be applied without additional assumptions on the latent space. This was achieved by an approach that tests the model’s ability to predict held out neurons. Without a good approximation of the latent structure underlying the neuronal population activity, this held-out prediction task is incredibly difficult to solve.
> >
> > > 7. Relation to prior work (Neurodata without borders, DANDI)
> >
> > The benchmark organizes several datasets by formatting them using the Neurodata Without Borders (NWB) standard and making them available on DANDI servers. We thank the reviewer for pointing out the risk of confusing our benchmark datasets with others that are publicly available on DANDI from the same labs that contributed to our benchmark. Our datasets contain distinct experimental sessions that are not otherwise available publicly, to prevent public access to our private evaluation data. We clarify this in the opening to Sec. 3.
> >
> > [1] Empirical models of spiking in neural populations. Macke et al. NeurIPS, 2011.

---

### Author Response · Authors · 2021-09-30
**Summary response to reviewers**

We thank the reviewers for their time and thoughtful critiques. We are pleased that the reviewers found our work well-motivated (Ty7Z, iRhp) and thorough in its use of several metrics and baselines (Ty7Z, iRhp), particularly with the diversity of datasets (bqKG) and secure challenge-style evaluation of unsupervised metrics (epMK, Ty7Z, iRhp).

Reviewers Ty7Z and iRhP were both very enthusiastic about the work, and their primary concern was the lack of statistical comparisons or confidence estimates in our model comparisons. This is an excellent point. Following the reviewers’ suggestions, we now 1) re-run baseline models multiple times with different random seeds, and 2) compare baseline models using bootstrap analyses. These analyses demonstrate the consistency of the LVMs and the robustness of our metrics.

Reviewer epMK and bqKG raised a concern that we may be neglecting supervised latent evaluation methods, which are more conventional in other ML domains. This is an important design choice to clarify. We describe in our work why supervised evaluation, which prescribes particular goals for the structure of latent space, is not generally appropriate for neural data LVMs. We thus prioritized the more generally applicable metric of co-smoothing, an unsupervised metric analogous to image inpainting in vision tasks. In recognition of the value that direct evaluation of latents provides, we provided, to the extent possible in our datasets, direct behavioral decoding on latents as a secondary metric.

Additionally, reviewers Ty7Z and bqKG would like to address how our application of these datasets differ from previous datasets used for LVM evaluation. To clarify this point, we have added a new supplementary table (Section A.2) that demonstrates the history of LVM development in ML conferences, and the lack of standardization around datasets and metrics in this effort.

We summarize additional changes made to our manuscript here:
- Added exposition on spiking activity and data pre-processing methods (epMK).
- Added an ethics discussion (Ty7Z).

We address reviewer concerns point-by-point in the detailed individual responses.

---

### Decision · Program_Chairs · 2021-10-11

**Decision:**

Accept

**Comment:**

The authors are proposing a dataset of spiking neural data recorded primarily from the motor areas of the primate brain while the primates are completing various reaching tasks. The neural recordings are also supplemented with hand kinematics data for the ground truth behavioural outcome of the neural activity. The authors have put together a nice evaluation challenge with private heldout data that will never be accessible to the users of their datasets, but which will be used privately on their servers for evaluating the submitted models. The dataset/leaderboard/competition setup can be a great way to push forward the developments of more accurate unsupervised models of spike activity.

The reviews are very mixed (4, 7, 5, 7) and all reviewers have quite low confidence. The main drawbacks brought up by the reviewers are:

Wrong/too-broad use of the term Latent Variable Model (LVM)
Lack of more traditional to the ML literature evaluation measures for the learnt latent spaces
Lack of statistical analysis of the results - are there statistically meaningful differences?
Is it fair for the competition to allow users access to the behavioural data - what if some models use this supervision signals while others don't?
The authors have addressed all of these points reasonably in my opinion during the discussion period. They have agreed to change how they refer to the baselines (not LVMs), they have added statistical tests, they have added a fair discussion on how the more traditional analysis of the learnt latents is not applicable in their use case, and how access to the behavioural data does not help these models pass all of their evaluation tests.

Having read the paper myself, and coming from the neuroscience background, I believe that the paper merits publication. It will be a great resource to encourage competition in developing good unsupervised learning methods for spiking neural data (which is being collected in more and more volume with the emergence of the better large scale recording methods), and gaining a better understanding of the latent dynamics in these populations of neurons. For this reason, I recommend acceptance.